# Enhanced Biodegradation/Photodegradation of Organophosphorus Fire Retardant Using an Integrated Method of Modified Pharmacophore Model with Molecular Dynamics and Polarizable Continuum Model

**DOI:** 10.3390/polym12081672

**Published:** 2020-07-27

**Authors:** Jiawen Yang, Qing Li, Yu Li

**Affiliations:** 1The Moe Key Laboratory of Resources and Environmental Systems Optimization, North China Electric Power University, Beijing 102206, China; yangjiawen236@163.com (J.Y.); lq0226@outlook.com (Q.L.); 2College of Environmental Science and Engineering, North China Electric Power University, Beijing 102206, China

**Keywords:** organophosphorus fire retardant, biodegradation, photodegradation, normalization method, 3D-QSAR pharmacophore model, molecular dynamics, polarizable continuum model

## Abstract

A comprehensive 3D-quantitative structure–activity relationship (QSAR) pharmacophore model was constructed using the values of comprehensive biodegradation/photodegradation effects of 17 organophosphorus flame retardants (OPFRs) evaluated by a normalization method to modify OPFRs with high biodegradation/photodegradation, taking tris(chloro-isopropyl) phosphate (TCPP), tris(2-chloroethyl) phosphate (TCEP) and tris(1-chloro-2-propyl) phosphate (TCIPP)—which occur frequently in the environment, and are the most difficult to degrade as target molecules. OPFR-derivative molecules TCPP–OH shows the highest improvement in biodegradation and photodegradation (55.48% and 46.37%, respectively). On simulating the biodegradation path and photodegradation path, it is found that the energy barrier of TCPP–OH for phosphate bond cleavage is reduced by 15.73% and 52.52% compared to TCPP after modification, respectively. Finally, in order to further significantly improve its biodegradability and photodegradation, the efficiency enhancement in the biodegradation and photodegradation of TCPP–OH are analyzed under the simulated environment by molecular dynamics and polarizable continuum model, respectively. The results of molecular dynamics show that the biodegradation efficiency of the TCPP–OH increased by 75.52% compared to TCPP. The UV spectral transition energy (4.07 eV) of TCPP–OH under the influence of hydrogen peroxide solvation effect is 44.23% lower than the actual transition energy (7.29 eV) of TCPP.

## 1. Introduction

The plasticity and heat resistance of plastic products are often enhanced by the addition of plasticizers and flame retardants during synthesis. The gradual elimination of brominated flame retardants (BFRs) led to its replacement by organophosphorus flame retardants (OPFRs), whose usage increased from about 3 × 10^5^ to 1 × 10^6^ t [1]. Based on their usage, OPFRs are divided into two types, namely, additive and reactive. The additive type adds OPFRs to polymer materials through physical mixing rather than chemical bonding and is thus exposed to the surroundings through volatilization, leaching, abrasion and dissolution [2]. The additive OPFRs volatilize easily from furniture and textiles to indoor air. The reactive OPFRs are chemically bonded to the polymer. Thus, they are stable, and possess a persistent plasticizing effect. Although non-volatile, reactive OPFRs from broken building materials and microplastics can flow into waste water treatment plants (WWTPs), rivers and oceans. The extensive use of OPFRs is likely to increase environmental pollution [3,4].

The frequent detection and persistence of OPFRs in the environment suggest that human beings can be exposed to OPFRs through different ways such as ingestion (inhalation) of indoor dust and consumption of contaminated food or drinking water [5,6,7]. Long-term exposure and accumulation of OPFRs in animals and humans may lead to adverse reactions, including renal toxicity, neurotoxicity, reproductive toxicity, carcinogenicity and endocrine disruption [8,9,10,11]. Lipophilic OPFRs are bioaccumulative, as seen from their detection in human hair, nails, urine and breast milk [12,13,14,15]. In addition, chlorinated OPFRs such as tris(2-chloroethyl) phosphate (TCEP), tris(1-chloro-2-propyl) phosphate (TCIPP) and tris(1, 3-dichloro-2-propyl) phosphate (TDCPP) were found to be neurotoxic and carcinogenic [16,17].

Considering the harmful effects of OPFRs, it is necessary to develop a reasonable method for its removal from the environment. Currently, the mainstream methods for the elimination of OPFRs include biodegradation activated sludge and phototransformation. Cristale et al. showed the degradation rate of OPFRs upon the conventional activated sludge treatment of WWTPs was just 49%, with 1% deposited in the sludge, and up to 50% discharged into freshwater or marine environment through sewage treatment plants [18]. Chlorinated OPFRs like TCEP, TCIPP, TDCPP, etc. are the most difficult compounds to undergo biodegradation. The traditional activated sludge process has a high removal rate (57–86) for alkyl and aryl OPFRs, while there is no remarkable degradation effect on chlorinated OPFRs. A process designed by the University of Cape Town was able to remove only 12.3% TCEP and 11.8% tris(chloro-isopropyl)phosphate (TCPP) [19]. TCPP is the most prevalent emerging pollutant in the European Union wastewater, accounting for about 80% of the European chlorinated OPFRs, with TCPP concentrations in surface water, groundwater and drinking water rising to 24 mg L^−1^ [5,20,21]. In a study, effluent samples were collected from the WWTPs in seven cities located in four different countries of Europe. Two chlorinated OPFRs—namely, TCPP and TCEP—were detected in most of the samples [22], indicating that the degradation of OPFRs by the sewage treatment plants in many European countries is not effective enough.

Given the insufficient degradation, researchers optimized the external conditions for the biodegradation/photodegradation of OPFRs. Ozonation and ultraviolet (UV)/H_2_O_2_ processes were implemented into the existing effluent treatment techniques of WWTPs to transform OPFRs by phototransformation. The alkyl OPFRs, namely tris(2-butoxyethyl) phosphate (TBOEP) and tributyl phosphate (TNBP) in sewage could be effectively eliminated by O_3_ and UV/H_2_O_2_, but the degradation of chlorinated OPFRs (TCEP, TCIPP and TDCPP) was not effective enough [18,22,23]. Therefore, this study uses chlorinated OPFRs such as TCPP, TCEP and TCIPP as target molecules to develop a new method for the construction of a comprehensive effect model that takes into account the biodegradation and photodegradation of OPFRs, based on a standardized method. The comprehensive effect model is used to perform molecular modifications for the development of environment-friendly OPFRs having high biodegradation/photodegradation efficiency, which improves the environmental removal rate. Considering the external conditions of biodegradation and photodegradation of target molecules in the real environment, molecular dynamics simulations assisted with Taguchi experimental design and UV spectrum calculation based on the solvation effect by density functional theory (DFT) are used for further enhancement of biodegradation/photodegradation efficiency of OPFRs derivatives. Thus, the degradation efficiency of OPFRs and their new designed derivatives in the real environment are significantly improved so as to account for the limitations of environmental removal and provide a new idea and method for the degradation of environment-friendly organophosphorus flame retardant materials.

## 2. Materials and Methods

### 2.1. Calculation of Photodegradation and Functional Properties of OPFR Molecules—Density Functional Theory (DFT) Method

The time-dependent density functional theory (TD-DFT) was used to calculate UV spectral information of OPFR molecules. First, the TD-DFT B3LYP (a calculation method of hybrid functionals) calculations were performed at the 6–31 g (d) basis set level using Gaussian 09 software (Gaussian, Inc., PA, America) to optimize the ground-state molecules. The UV spectra of OPFRs were calculated under optimal conditions, and the corresponding wavelength (λ) and energy required for the transition of 22 OPFR homologs to their first excited state were obtained. The energy required for the first excited state transition of OPFRs from the UV spectrum was used to characterize the photodegradation ability of these molecules. As energy decreases, the photodegradation efficiency increases [24]. The energies required for the first excited state transition of 22 OPFRs from the UV spectrum are shown in Table 1.

In order to further significantly improve the photodegradation of OPFR derivatives, the efficiency enhancement in the photodegradation of TCPP–OH (which showed the highest improvement in photodegradation) was analyzed under the simulated environment by polarizable continuum model (PCM). The UV spectral information of TCPP and TCPP–OH in hydrogen peroxide solvent that generated hydroxyl radicals was calculated using the PCM based on the self-consistent reaction field (SCRF) theory at B3LYP/6–31 g (d) basis set level. The information from the UV spectrum was used to analyze the enhancement of photodegradation under the solvation effect of OPFRs and their derivatives [24,25]. The Gibbs free energy (ΔG) and reaction energy barrier (ΔE) of OPFRs and their derivative molecules in biodegradation and photodegradation reactions at B3LYP/6–31 g (d) basis set level were computed, as shown in Formulas (1) and (2).
(1)ΔG=ΣG(product)−ΣG(reactant)
(2)ΔG=E(TS)−ΣE(reactant)
where TS in Formula (2) represents the transient state. The TS of the substitution reactions have a single virtual frequency and the reaction paths are validated by the intrinsic reaction coordinates (IRC).

The functional characteristics of OPFRs before and after molecular modification were evaluated by the energy gap (eV), total energy (a.u.), positive frequency value and fire resistance. The energy gap value refers to the energy difference between the highest occupied molecular orbital (HOMO) and lowest unoccupied molecular orbital (LUMO). The conductivity decreased with an increase in the energy gap [26]. The energy value and positive frequency value represent the stability of OPFRs in the environment [27]. The B3LYP/6–31 g (d) basis set level was used to calculate the molecular energy gap, energy value and positive frequency value of OPFRs before and after molecular modification.

The gas products formed by the pyrolysis of OPFRs contain free radical PO• which could capture H• and •OH free radicals from the flame and thus reduce their concentration. H• and •OH could inhibit the combustion chain reaction, and eventually slows down or terminates the combustion process. The P–O bond cleavage is the core of the flame retardant reaction; hence the flame retardant efficiency of OPFR is related to the strength of the P–O bond. Therefore, the P–O bond dissociation enthalpies of OPFRs were selected as the parameters for the evaluation of flame retardancy. As the P–O bond dissociation enthalpy of OPFR decreased, there was an enhancement in the flame retardancy [28]. The dissociation enthalpy was calculated using the following formula:(3)$$\mathrm{Dissociation}\;\mathrm{enthalpy}\left(P-O\right)=\left(O\right)+H_{298}^{0}\left(P\right)-H_{298}^{0}\left(\mathrm{OP}\right)$$
(4)$$H_{298}^{0}=E+\Delta \mathrm{ZPE}+{\Delta H}_{\mathrm{trans}}+{\Delta H}_{\mathrm{rot}}+{\Delta H}_{\mathrm{vib}}+\mathrm{OT}$$
where ZPE is the zero-point energy, ΔH_trans_, ΔH_rot_ and ΔH_vib_, respectively represent the contribution of translation, rotation and vibration to energy, and T(K) is the specific temperature.

### 2.2. Characterization Method for the Comprehensive Biodegradation/Photodegradation Effect Values of OPFR Molecules—Normalization Method

The actual values of biodegradation and photodegradation effects of OPFR molecules were converted into dimensionless efficacy coefficients, *R*_*S*1,*i*_ and *R*_*S*2,*i*_, respectively, by using Formula (5) of the normalization method. The purpose of this study was to improve the biodegradability and photodegradation of OPFR molecules without distinction. Thus, the biodegradation and photodegradation weights (*w*) were selected to be 50%. Based on the set index weight, the comprehensive effect evaluation values represented by C (in Table 1) of biodegradation/photodegradation of OPFRs were calculated using Formula (6). These values were used to evaluate the biodegradation/photodegradation effects of OPFR molecules.
(5)$$R_{S,j}=\frac{min\left\{x_{i}^{*},T\right\}}{max\left\{x_{i}^{*},T\right\}}$$
where *T* represents the target or ideal value in the given data, which is taken as the median of the data, $x^{*}=\left\{x_{1}^{*},x_{2}^{*},\ldots ,x_{n}^{*}\right\}$ represents the single effect value for the binding free energy of OPFR molecule binding to its degrading enzyme and $x_{i}^{*}$ is the binding free energy value of the *i*th OPFR molecular binding to its degrading enzyme.
(6)$$C=wa\times R_{S1,i}+wa\times R_{S2,i}$$
where $R_{S1,i}$ and $R_{S2,i}$ represent the dimensionless values for the biodegradation and photodegradation effects of OPFR molecules, respectively, which are transformed by the normalization method. As the comprehensive parameter decreases, the biodegradation/photodegradation efficiency increases.

### 2.3. Enhancement of Degradation Efficiency for OPFR-Derivative Molecules—Molecular Dynamics Method Assisted with Taguchi Experimental Design

In this study, the structure of proteases was derived from the protein data bank (PDB). The binding free energy of OPFRs and degrading enzyme (the ID of this degrading enzyme in PDB is 5HRM) in *Sphingobium* sp. strain TCM1 represent the biodegradability of OPFRs [29]. Using Gaussian 09 software, DFT B3LYP calculations at 6–31 g (d) basic set level were performed to optimize the structure of OPFR molecules and their derivatives. The binding free energy (ΔG_bind_) of 22 OPFR molecules and receptor proteins were calculated under optimal conditions; the results are shown in Table 1. Studies have shown that stronger binding free energy between enzyme (PDB ID: 5HRM) and target molecule makes the biodegradation reaction of OPFRs more favorable [29]. There is stronger interaction between the degrading enzyme and the target molecule as the binding energy decreases [30]. The binding free energy between the enzyme (PDB ID: 5HRM) and OPFRs was calculated based on the molecular dynamic simulation module of the Gromacs software in the Dell PowerEdge R7425 server (Berendsen Laboratory, Göttingen University, Göttingen, Germany). The composite system of enzyme (PDB ID: 5HRM) and OPFR molecules was placed in a cube with a side length of 8.3 nm. The GROMOS96 43A1 force field was used for molecular constraint, and Na^+^ was added to neutralize the system charge. The above composite system was set as a group and the energy minimization simulations based on the steepest gradient method were performed with the simulated steps set to 100,000. The heat bath and pressure simulation times of the composite system were set to 100 ps, with a constant standard atmospheric pressure of 1 bar, and the dynamic simulation calculation time of each level was set to 200 ps [30].

In order to further significantly improve the biodegradability of OPFR derivatives, the efficiency enhancement in the biodegradation of TCPP–OH (which showed the highest improvement in biodegradation) was analyzed. The molecular dynamics method assisted with Taguchi experimental design was used to improve the biodegradation efficiency of the derivative TCPP–OH molecule by simulating the process of biodegradation in the real environment. The binding free energy between TCPP–OH and degrading enzyme (PDB ID: 5HRM) that represents the biodegradability of TCPP–OH was used to determine the optimum external conditions that promote the biodegradation of the derivative molecule. Former studies have shown that the pH(A), temperature (B), carbon source methanol (C), acetic acid (D), oxygen promoter concentration (E), voltage gradient (F) and surfactant concentration (G) affect the biodegradation of OPFRs and other organic compounds [18,31,32,33]. As a special orthogonal experimental method, Taguchi experimental design is a special orthogonal experimental method as fewer experiments are required to analyze a large number of variables [34]. In this study, the Taguchi orthogonal experimental design of L_26_ (3^7^) was conducted by selecting the external conditions that promote the biodegradation of OPFRs as the variables to generate the orthogonal experimental table and the appropriate conditions of each variable as the experimental level. Molecular dynamic simulation for the biodegradation of OPFRs was performed according to the generated orthogonal experimental table. The binding free energy between the degrading enzyme (PDB ID: 5HRM) and the derivative TCPP–OH was calculated based on the molecular dynamic simulation module of the Gromacs software. The composite system of degrading enzyme (PDB ID: 5HRM) and TCPP–OH-derivative molecules was placed in a cube of 15nm side length, with the GROMOS96 43A1 force field used for molecular constraint and Na^+^ added to neutralize the charge of the system. The above composite system was set as a group and the energy minimization simulation based on the steepest gradient method was performed. The pressure was set to a constant standard atmospheric pressure of 1 bar. According to the binding free energy, the optimal external environmental conditions that efficiently promote the degradation reaction of TCPP–OH with degrading enzyme were determined. Moreover, the biodegradation process of the derivative molecule TCPP–OH was simulated under the optimal external environmental condition to evaluate the biodegradation ability of the new derivative.

## 3. Results and Discussion

### 3.1. Construction and Reliability Test for Biodegradation and Photodegradation Pharmacophore Models of OPFRs

This study used Discovery Studio 4.0 software (BIOVIA™, Vélizy-Villacoublay, France) to establish a 3D–QSAR pharmacophore model using the quantitative values for the comprehensive biodegradation/photodegradation effects of OPFR molecules as dependent variables and molecular structure parameters as independent variables. A total of 17 OPFR (TDCPP, tri(n–propyl) phosphate (TnPP), cresyl diphenyl phosphate (CDPP), TCEP, tris(p-*t*-butylphenyl) phosphate (TBPP), tripropyl phosphate (TPrP), triisobutyl phosphate (TiBP), TCIPP, butyl diphenyl phosphate (BdPhP), triphenyl phosphate (TPHP), isodecyl diphenyl phosphate (IDPP), trimethyl phosphate (TMP), tripentyl phosphate (TPeP), tris(2-ethylhexyl) phosphate (TEHP), 2-ethylhexyl diphenyl phosphate (EHDPP), triethyl phosphate (TEP), TCPP) molecules were randomly selected as the training set to obtain the pharmacophore model for the comprehensive biodegradation/photodegradation effects by Hypo-Gen and statistical data (Table 2). The test set consisted of five OPFR molecules for the verification of the comprehensive effect pharmacophore model (Table 3).

As shown in Table 2, Hypo C1 had the best evaluation scores among the nine pharmacophore models constructed with Hypo-Gen for the comprehensive biodegradation/photodegradation effects of OPFRs. It had the smallest total cost (75.55) and root-mean-squared (RMS) error (0.24), with the maximum R^2^ of 0.90 (>0.7), (Hypo C2, C4, C6 and C8 were not considered because both the RMS and R^2^ values were 0). Moreover, the Hypo C1 pharmacophore model had a configuration value of 16.75 (<17), indicating the best predictions for the comprehensive biodegradation/photodegradation effects of OPFRs. Thus, the significance of the model was established.

The reliability and predictability of the molecular comprehensive biodegradation/photodegradation effect of OPFRs by Hypo C1 were evaluated using a test set. It was found that there was homogeneity between Hypo C1 and the five OPFR molecules in the test set with error values of less than 2 (Table 3), which was within the allowable error range. This indicated the stable predictability of the model in comparison with the OPFR molecules in the training set. Therefore, the force field information displayed by the pharmacophore model Hypo C1 for comprehensive biodegradation/photodegradation effects was used to modify the target OPFR molecules.

Moreover, the reliability of the molecular modification of OPFRs based on the comprehensive effect pharmacophore model should be ensured by the employment of single-effect pharmacophore models for biodegradation and photodegradation and prediction of the activity data (binding free energy and first excited state transition energy) of each model. This could be used to obtain OPFR-derivative molecules with improved properties of biodegradation, photodegradation and comprehensive biodegradation/photodegradation effects depicted by the evaluation values.

By selecting ten OPFR molecules randomly as the training set and three OPFRs as the test set, the optimal pharmacophore model HypoB1 for the biodegradation effect of OPFRs was established, with ΔG_bind_ as the dependent variable and molecular structure parameter as the independent variable. Similarly, the optimal pharmacophore model Hypo P1 for the photodegradation effect of OPFRs was obtained. The parameters of the model are listed in Table 4.

HypoB1 and HypoP1 had the smallest total cost (48.17 and 50.23), with minimum RMS error (0.03 and 2.20) and maximum R^2^ values greater than 0.7 for both models. The configuration values of HypoB1 and HypoP1 were less than 17, indicating high predictability of the pharmacophore models HypoB1 and HypoP1 for the biodegradation and photodegradation effects of OPFRs.

### 3.2. Molecular Modification of OPFRs with High Biologic/Photodegradation Efficiency

Three chlorinated OPFRs molecules, namely, TCPP, TCEP and TCIPP, which are difficult to undergo biodegradation and photodegradation, were selected to perform molecular modification for the enhancement of biodegradation/photodegradation efficiency. The Hypo C1 pharmacophore model was used for the analysis of force fields generated by TCPP, TCEP and TCIPP molecules, to determine the modified sites and substituent groups in OPFR molecules.

As seen from Figure 1, TCPP, TCEP and TCIPP were affected by hydrogen bond acceptor group (green) and hydrophobic group (blue). It indicated that the introduction of such groups improves the activity parameters of the molecule. Moreover, the comprehensive biologic/photodegradation efficiency enhances as the comprehensive effect evaluation value lowers. Thus, the objective of this study was to design OPFR derivatives with low biodegradation/photodegradation effect evaluation value. This required the introduction of hydrogen bond donors and hydrophilic groups at key sites of the target molecules.

As shown in Figure 2, the hydrophobic groups (marked in blue) acted on the sites 4′, 6′ and 7′ of TCPP, TCEP and TCIPP molecules, respectively. Thus, hydrophilic groups should be introduced at site 4′ of TCPP, site 6′ of TCEP and site 7′ of TCIPP to reduce the comprehensive effect evaluation value of target molecules. In addition to site 7′, hydrogen bond receptors were present at site 4′’ of TCIPP. Thus, hydrogen bond donors should be introduced at this site to reduce the comprehensive effect evaluation value of TCIPP.

Nine hydrophilic groups were selected as substituent groups, including hydroxyl (–OH), amino (–NH_2_), carboxyl (–COOH), amide (–CONH_2_), aldehyde (–CHO), phosphate (–PO_3_H_2_), acylamino, (–CONH_2_),–COOCH_3_ and sulfonic groups (–SO_3_H), among which–OH,–NH_2_,–COOH,–CONH_2_ and–CHO were both hydrophilic and hydrogen bond donors. Out of these, six hydrogen bond donor groups were selected, including fluoro (–F), chloro (–Cl), bromo (–Br), nitro (–NO_2_), carbonyl (–COCH_3_) and methoxy (–OCH_3_).

The pharmacophore model Hypo C1 was used to conduct monosubstituted reactions on TCPP and TCEP molecules, while mono- and disubstituted reactions on TCIPP molecules. A total of 58-derivative molecules were designed (38 single monosubstituted and 20 disubstituted derivative molecules), among which the comprehensive biodegradation/photodegradation effect evaluation values for 19 OPFR derivatives were reduced. The predicted results are shown in Table 5.

### 3.3. Prediction of Biodegradation and Photodegradation for OPFR-Derivative Molecules

The established pharmacophore models for the comprehensive and single biodegradation and single-effect photodegradation effects were used for the prediction of the activity of OPFR-derivative molecules. The predicted values and degree of enhancement are shown in Table 5.

Upon molecular modification, the comprehensive effect evaluation values of 18 derivatives of the target TCPP and TCEP molecules were reduced to a great extent from 6.36% to 44.06%. Based on the predicted results for biodegradation and photodegradation effect models of OPFRs, the evaluation values for the biodegradation (−logLC_50_) of the 18 derivatives of TCPP and TCEP molecules increased, and those of photodegradation decreased, which were consistent with the trend of biodegradation/photodegradation effect evaluation value. The weights of both biodegradability and photo-degradability were set to 50% for the calculation of comprehensive biodegradation/photodegradation effect evaluation values of OPFRs. Based on the predicted results (Table 5), the 14 derivative molecules obtained showed a reduction in the individual biodegradation and photodegradation effects to the extent of 50%:50%. This was observed in TCPP–OH, TCPP–CH_2_OH, TCPP–COOH, TCPP–CONH_2_, TCPP–CHO, TCPP–PO_3_H_2_, TCPP–SO_3_H, TCEP–OH, TCEP–CH_2_OH, TCEP–COOH, TCEP–CONH_2_, TCEP–PO_3_H_2_, TCEP–COOCH_3_ and TCEP–SO_3_H. The results indicate that the weight setting for the pharmacophore model of the comprehensive biodegradation/photodegradation effects of OPFRs was representative, with practical significance.

The target TCIPP molecule individually modified with nine groups (–OH,–CH_2_OH,–NH_2_,–COOH,–CONH_2_,–CHO,–PO_3_H_2_,–COOCH_3_,–SO_3_H) were selected for mono– and di–substitution reactions for the formation of 40 TCIPP-derivative molecules. However, the predicted evaluation values for the comprehensive biodegradation/photodegradation effects of 40 TCIPP-derivative molecules showed an upward trend from –1.61% to 112.29%. Only three derivative molecules, namely TCIPP–NH_2_, TCIPP–CONH_2_ and TCIPP–F show a downward trend in the predicted comprehensive evaluation values, with a reduction of 5.50%, 2.00% and 22.00%, respectively. However, the overall increase in the evaluation value suggested that the biodegradation/photodegradation efficiency of TCIPP could not be improved by molecular modification. Thus, TCPP and TCEP molecules were considered for the development of OPFRs with low comprehensive biodegradation/photodegradation effect evaluation values. Therefore, a total of 14 derivatives, namely, TCPP–OH, TCPP–CH_2_OH, TCPP–COOH, TCPP–CONH_2_, TCPP–CHO, TCPP–PO_3_H_2_, TCPP–SO_3_H, TCEP–OH, TCEP–CH_2_OH, TCEP–COOH, TCEP–CONH_2_, TCEP–PO_3_H_2_, TCEP–COOCH_3_, TCEP–SO_3_H were found to have enhanced effects, based on the predictions of biodegradability and photo-degradability for the comprehensive biodegradation/photodegradation effect model and single effect models for OPFR-derivative molecules.

### 3.4. Practicality Evaluation of OPFRs Derivatives Molecular

The flame retardancy, energy value, energy gap value and frequency of 14 OPFR-derivative molecules were calculated, among which seven environment-friendly derivative molecules with good functional considerations are finally selected. The results are shown in Table 6.

According to the calculated results by Gaussian in Table 6 for 14 TCPP and TCEP-derivative molecules, 11 molecules showed an increase in the energy gap from 0.19% to 34.04%. The energy gap values of TCPP–OH, TCPP–CONH_2_ and TCPP–SO_3_H decreased slightly to the extent of less than 5%. The energy gap value was used to represent insulation. Insulation increased with an increase in the energy gap value. The change in energy gap values of the above derivative molecules showed no significant decrease in insulation. In addition, the energy of the 14 derivative molecules decreased to different degrees compared to that of TCPP and TCEP. The frequencies of the molecules were found to be positive, indicating the high stability for the structures of TCPP and TCEP-derivative molecules.

The O–P bond dissociation enthalpy was used to characterize the flame retardant properties of OPFRs and the new derivative molecules. Among the new derivative molecules with high biodegradability and photo–degradability, seven molecules, namely, TCPP–OH, TCPP–CONH_2_, TCPP–CHO, TCPP–PO_3_H_2_, TCEP–PO_3_H_2_, TCEP–COOCH_3_ and TCEP–SO_3_H show an increase in the flame retardancy by 7.15%, 8.81%, 4.33%, 10.58%, 12.56%, 6.51% and 0.76%, respectively, which meet the requirements for environment-friendly OPFRs.

### 3.5. Mechanistic Analysis for the Biodegradation and Photodegradation of OPFR-Derivative Molecules

It can be seen from Table 6 that TCPP–OH was the derivative molecule with the greatest improvement in biodegradation and photodegradation among the seven environment-friendly OPFRs. The biodegradation and photodegradation efficiencies of the derivative TCPP–OH increased by 55.48% and 46.37%, respectively. Thus, further investigation for the improvement of biodegradation and photodegradation efficiencies before and after TCPP molecular modification was performed using TCPP–OH as the target derivative molecule.

#### 3.5.1. Mechanistic Analysis for The Enhancement of Biodegradation for OPFR-Derivative Molecules

Studies on the biodegradation path of OPFRs have shown that *Sphingobium* sp. strains play a major role in the degradation process as tributyl phosphate (TBP) is hydrolyzed under the effect of the bacterium. After the cleavage of one butyl side chain forms an intermediate product di-n-butyl phosphate (DnBP), a second butyl group is broken and further degraded to generate mono-n-butyl phosphate (MnBP). Finally, the cleavage of three butyl chains is observed, which leads to the degradation of TBP to the phosphate group Pi [35]. Previous studies reported that TCEP and TPP are hydrolyzed under the action of the enzyme in the *Sphingobium* sp. strain TCM1. The successive cleavage of the phosphate ester bond is a common step in the biodegradation of OPFRs such as TCEP, TBP and TPP [15,29,36]. The biodegradation path of TCPP and its derivative molecule TCPP–OH was simulated in accordance with the biodegradation path of TBP, which led to the greatest improvement in biodegradation and photodegradation and deduction of microbial degradation products (Figure 3). The difficulty of biodegradation before and after the molecular modification of TCPP was analyzed by calculating the reaction barrier.

In accordance with the biodegradation pathway of TBP molecules catalyzed by *Sphingobium* sp., the hydrolysis of the phosphate ester bond in TCPP and TCPP–OH generated products I, II and III. The energy barrier of the biodegradation reaction of TCPP and TCPP–OH were calculated and marked for each step in the degradation reaction, as shown in Figure 3. As the reaction energy barrier lowers, the probability of the biodegradation reaction increased [24]. As seen in Figure 3, different side chains of TCPP and TCPP–OH were cleaved to generate product I. The hydrolysis of the first phosphate bond of TCPP generated 2-chloropropanol, while the hydrolysis of the phosphate bond of TCPP–OH molecule produces HO–OH, which gets removed. Therefore, the energy barrier for this step in the side-chain fracture reaction of TCPP and TCPP–OH was different. Compared to the reaction barrier (ΔE = 249.4895 kcal·mol^−1^) of TCPP molecule before modification, the energy barrier value (ΔE = 210.2360 kcal·mol^−1^) of the newly designed derivative molecule TCPP–OH was reduced by 39.2535 kcal mol^−1^. Thus, the total energy barrier for the biodegradation of TCPP–OH was reduced by 15.73%, relative to the TCPP molecule. The results show that the modified derivative molecule TCPP–OH had effectively reduced the reaction energy barrier for the phosphate ester bond cleavage process of biodegradation, thus making the molecule more degradable.

#### 3.5.2. Mechanistic Analysis for the Enhancement of Photodegradation for OPFR-Derivative Molecules

Compared to traditional biodegradation and photodegradation processes, the UV/H_2_O_2_ photolysis in advanced oxidation processed (AOPs) was found to improve the removal rate of chlorinated OPFRs to the maximum extent [21,37]. The UV/H_2_O_2_ process was the generation of hydroxyl radical (•OH) through the photolysis of H_2_O_2_. The •OH radical was highly reactive and corrodes most of the organic molecules at a very high rate constant [38,39]. Other studies showed that the degradation kinetics of TCPP under visible light irradiation was slower than that under UV-visible irradiation, probably due to the less production of •OH radicals [1,14]. Moreover, •OH was proven to be the main active free radical in the aquatic environment, which degrades a variety of organic pollutants rapidly and effectively [40,41,42].

He et al. analyzed the kinetics and mechanism for the degradation of high concentration TCPP in the UV/H_2_O_2_ system and found that TCPP was transformed into several products of hydroxylation and dechlorination as the reaction progressed and determined the intermediates for the reaction [21]. Further, the degradation pathway of TCPP was inferred using the nature of the intermediated. TCPP had four degradation pathways. Route 1 involved the addition and extraction reactions of •OH to form 1-chloropropane-2-ol. Route 2 was the addition reaction of 1-chloropropane-2-alcohol with •OH under dehydration conditions and oxidation of the product. Route 3 included the extraction reaction of TCPP and addition of •OH radical for oxidation. Route 4 produced oxygen through the oxidation of H_2_O_2_ produced by the addition reaction of OH radicals. With reference to the photodegradation mechanism of TCPP in the UV/H_2_O_2_ system, this study simulated the photodegradation paths of the derivative TCPP–OH molecule before and after modification (Figure 4). The reaction energy barriers of the four main reactions of TCPP and TCPP–OH were calculated and marked at the corresponding positions, as shown in Figure 4.

From Figure 4, it can be observed that the reaction energy barriers for the four photodegradation paths of the derivative TCPP–OH molecule were lower than that of TCPP. This indicates that TCPP–OH was more easily degradable than the target TCPP molecule. Under the same reaction conditions, TCPP–OH had a higher degradation degree and efficiency. Moreover, •OH—being the most important active radical in the photodegradation of OPFRs—showed an active ability to participate in the reactions of TCPP in photodegradation paths 1, 2 and 4. In photodegradation path 3 of TCPP–OH, the O–OH bond on the side chain of TCPP–OH was broken to generate •OH radical and compound f (i.e., compound IX, the final product of TCPP path 3). The reaction energy barrier for the generation of •OH and compound f from TCPP–OH was calculated to be 190.8840 kcal mol^−1^, which was 211.1399 kcal mol^−1^ (52.52%) lower than that of compound IX (402.0239 kcal mol^−1^). The results show that the photodegradation reaction for the cleavage of the O–OH bond of the TCPP–OH molecule occurs preferentially and easily. Moreover, the active •OH radical—which promotes the photodegradation of OPFRs—was produced during the photodegradation reaction of TCPP–OH. Thus, OPFR molecules were more rapidly and effectively degradable.

### 3.6. Enhancement of Biodegradation and Photodegradation Efficiencies of New Derivative Molecule TCPP–OH under Simulated Environment

The biodegradability and photo–degradability of OPFRs and their derivatives are not only related to the molecular nature and structure, but also the external conditions in the real environment. Therefore, in addition to molecular modification, the external environmental conditions should also be considered to significantly improve the degradation efficiency of OPFR molecules in the real environment. Taking TCPP–OH as the research object, the molecular dynamic simulation method assisted with Taguchi experimental design was used to simulate and analyze the biodegradation process of TCPP–OH in the normal environment, aiming to further improve the efficiency of the process. The UV spectra of TCPP and TCPP–OH under the solvation effect were calculated by PCM using the SCRF theory. The photodegradation of TCPP and TCPP–OH in the solvent environment where can produce •OH was simulated to explore further and verify the enhanced photodegradation effect of OPFRs and their derivatives.

#### 3.6.1. Enhanced Biodegradability of TCPP–OH by Molecular Dynamic Simulation Assisted with Taguchi Experiment Design

The L_27_ (3^7^) Taguchi experimental design was used to establish the partial factor design for the derivative molecule TCPP–OH to improve the photodegradation efficiency and calculate the binding free energy values of TCPP–OH and degrading enzyme (PDB ID: 5HRM) under 27 experimental conditions. The 27 experimental conditions include factors of (A) pH, (B) temperature, (C) carbon source methanol, (D) acetic acid, (E) oxidation modification agent H_2_O_2_ concentration, (F) voltage gradient and (G) surfactant sodium dodecyl sulfate (SDS) concentration. According to the biodegradation treatment process of OPFRs and other organic pollutants, each factor was set to the appropriate levels such as low, medium and high [18,31,32,33]. The combination scheme of the seven factors and three levels is shown in Table 7. The optimal combination of external conditions that promote the biodegradation of the new derivative molecule TCPP–OH was determined by factorial analysis. The binding free energies found using the molecular dynamic simulation and combination scheme of the external conditions by the L_27_ (3^7^) Taguchi experiments of the TCPP–OH molecule are listed in Table 7. The biodegradation of TCPP and TCPP–OH were simulated by molecular dynamics under optimal conditions. The binding energies of degrading enzyme with the target molecule TCPP and the new derivative molecule TCPP–OH were calculated to investigate the enhancement of degradation capability for TCPP–OH in the real environment. The aim was to further improve the degradation of TCPP–OH molecules in the real environment by combining the effects of internal modification and external conditions.

As the mean value of binding free energy decreases, the binding affinity between TCPP–OH and degrading enzyme increases. Thus, the optimal external stimulus conditions by the combination of seven factors and three levels shown in Figure 5 are as follows: oxidation modification agent H_2_O_2_ concentration (200 mg/L), voltage gradient (1 V/cm), pH (6.5), temperature (303 k), carbon source methanol (500 mg/L), surfactant sodium dodecyl sulfate concentration (750 mg/L) and acetic acid (200 mg/L). In addition, different levels of factors A (pH), E (H_2_O_2_) and F (voltage gradient), which were shown to have a significant impact on the mean binding energy value of degrading enzyme and TCPP–OH, while the change in factor D (acetic acid) and G (surfactant sodium dodecyl sulfate) at a low, medium and high level, had less impact. The results are consistent with the ranking results calculated using the binding energy based on molecular dynamics in Table 8. The ranking of degrading enzyme degradation efficiency promoted by external conditions are listed in the descending order as follows: (E) oxidation modification agent H_2_O_2_ concentration, (F) voltage gradient, (A) pH, (B) temperature, (C) carbon source methanol, (G) concentration of the surfactant—SDS and (D) acetic acid. The effective operation of the biodegradation process in sewage treatment plants was due to the sufficiently dissolved oxygen in the sewage and activated sludge mixtures, which suspended the activated sludge within the tank to ensure a continuous supply.

Using the Taguchi experimental design analysis, it was found that the factor that significantly affects the degradation efficiency of degrading enzyme in TCPP, and that TCPP–OH was the concentration of oxygen promoter H_2_O_2_, which was ranked first among the external conditions. It was seen that the presence of H_2_O_2_ promotes the biodegradation and UV degradation of organic pollutants such as OPFRs. This is consistent with the results which state that the dissolved oxygen had a great influence on the biodegradation rates of OPFRs and other organic pollutants [15,21,35,37].

The binding energies of degrading enzyme with TCPP and TCPP–OH molecules were calculated based on molecular dynamic simulation under optimal conditions. The binding energies values were found to be 62.326 kJ/mol and−109.395 kJ/mol, respectively. The binding affinity between degrading enzyme and TCPP–OH increased by 75.52% compared to TCPP, indicating the high degradation efficiency of the derivative TCPP–OH molecule relative to the premodified TCPP, under optimal environmental conditions.

Therefore, the biodegradation simulation results of TCPP–OH (increased by 75.52%) were 20.04% higher than that of TCPP (increased by 55.48%) under optimal external conditions, as predicted by the biodegradation pharmacophore model (Figure 6). This indicates that the biodegradation efficiencies of OPFR derivatives were yet to be improved. This was done by combining the process of internal modification of the target OPFR molecules with the optimal selection of external conditions in the real environment. Thus, the biodegradation efficiencies of OPFRs were significantly promoted under molecular and external environmental conditions. This provided theoretical guidance for the significant improvement of biodegradation efficiencies of OPFRs and their derivatives.

#### 3.6.2. Enhanced Photodegradability of TCPP–OH by Solvation Effect Based on PCM

As seen from the earlier results of this study, the photodegradation mechanism shows that the O–OH bond cleavage of TCPP–OH occurs more preferentially and easily than that of TCPP. The active •OH radical that promotes the photodegradation of OPFRs were generated in the reaction process, thus leading to rapid and effective degradation of the molecules. Therefore, the UV spectra of TCPP and TCPP–OH under solvation effects were calculated using PCM by Gaussian 09 to investigate the effects of external environmental conditions on the UV degradation of TCPP–OH. The TCPP and TCPP–OH molecules were taken in hydrogen peroxide solvent to simulate the environment for the generation of •OH radicals. The energy required for the first UV excited state transition of the molecules before and after the solvation effect were compared to verify the enhancement of photodegradation of OPFRs and their derivatives in the simulated environment.

The photodegradation of the molecule was more feasible at low-UV spectral transition energy. Using hydrogen peroxide as the solvent, the minimum UV spectral transition energy for TCPP–OH was found to be 4.07 eV, and that of TCPP was 5.14 eV. The transition energy of TCPP without the solvation effect (as shown in Table 1) was found to be 7.29 eV. In addition, the UV-spectral transition energy of the TCPP–OH molecule without the solvation effect was calculated to be 5.47 eV. The UV spectral transition energy of TCPP–OH was reduced by 20.88% compared to that of TCPP molecules with the same solvation effect. This indicates that the photodegradation efficiency of TCPP–OH was still higher than TCPP under the same environment, due to its molecular properties and structure, which was in line with the predicted results for the photodegradation pharmacophore model. The UV transition energy of TCPP and TCPP–OH with the solvation effect was 29.52% and 25.74% lower than the same molecules without the solvation effect, respectively. It showed that the environment that produced •OH radicals stimulated by hydrogen peroxide reduced the UV-spectral transition energy during the photodegradation of TCPP and TCPP–OH molecules, which increased the feasibility of photodegradation. The •OH radical was an important factor affecting the photodegradation of OPFRs, as it increased the photodegradation efficiencies of TCPP and TCPP–OH, which is consistent with the results of Yuan et al. and Yu et al. [1,14,38]. The UV spectral transition energy (4.07 eV) of TCPP–OH in hydrogen peroxide was 44.23% lower than that of TCPP without the solvation effect. This indicated that the photodegradation efficiency could be further improved by molecular modification under the optimum external environment.

## 4. Conclusions

In this study, a 3D-QSAR pharmacophore model was constructed using the normalization method, which takes into account the evaluation value for comprehensive biodegradation/photodegradation effects and structural parameters of OPFRs. It was successfully applied to the molecular modification of environment-friendly OPFRs with high biodegradation/photodegradation efficiencies and functional properties, mitigating potential health risks to humans and other organisms.

In addition, it was found that the biodegradation/photodegradation of OPFR-derivative molecules can be further improved using molecular dynamics and PCM to adjust the external conditions, which simulate the degradation process. The results showed that the biodegradation and photodegradation efficiencies of OPFRs and their new designed derivative molecules in the real environment are significantly improved by the molecular modification method under optimum external environmental conditions. Thus, the limitations for the environmental removal of OPFRs and their new designed derivative molecules are resolvable at the source. In addition, a new idea and method for the degradation of environment-friendly substitutes are provided.

## Figures and Tables

**Figure 1 polymers-12-01672-f001:**
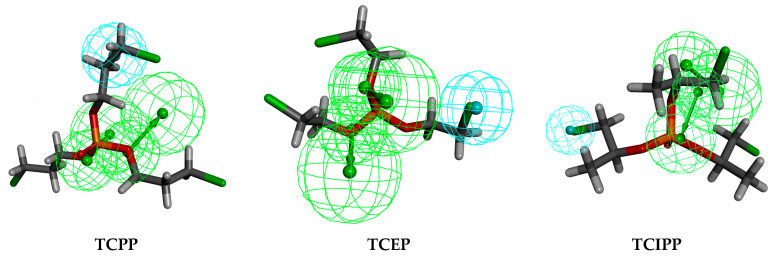
Three-dimensional congruent map of Hypo C1 and TCPP, TCEP and TCIPP.

**Figure 2 polymers-12-01672-f002:**
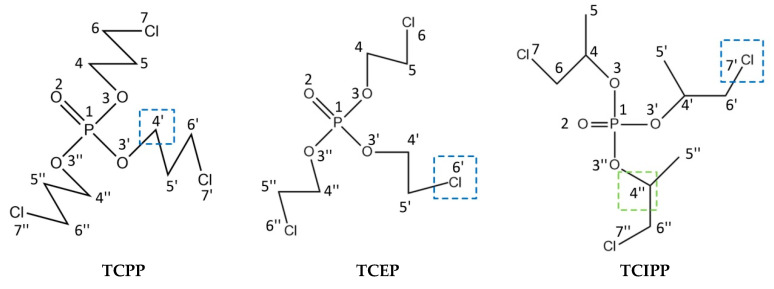
Schematic diagrams of the substitution positions affecting the comprehensive biodegradation effect values of TCPP, TCEP and TCIPP.

**Figure 3 polymers-12-01672-f003:**
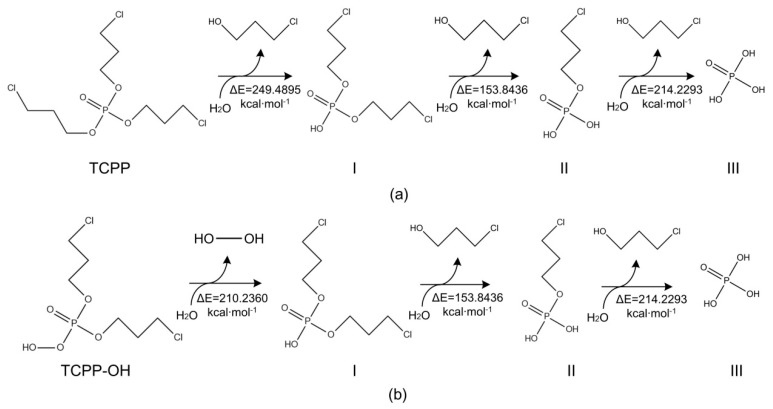
Proposed biodegradation pathway of TCPP before (**a**) and after modification (**b**).

**Figure 4 polymers-12-01672-f004:**
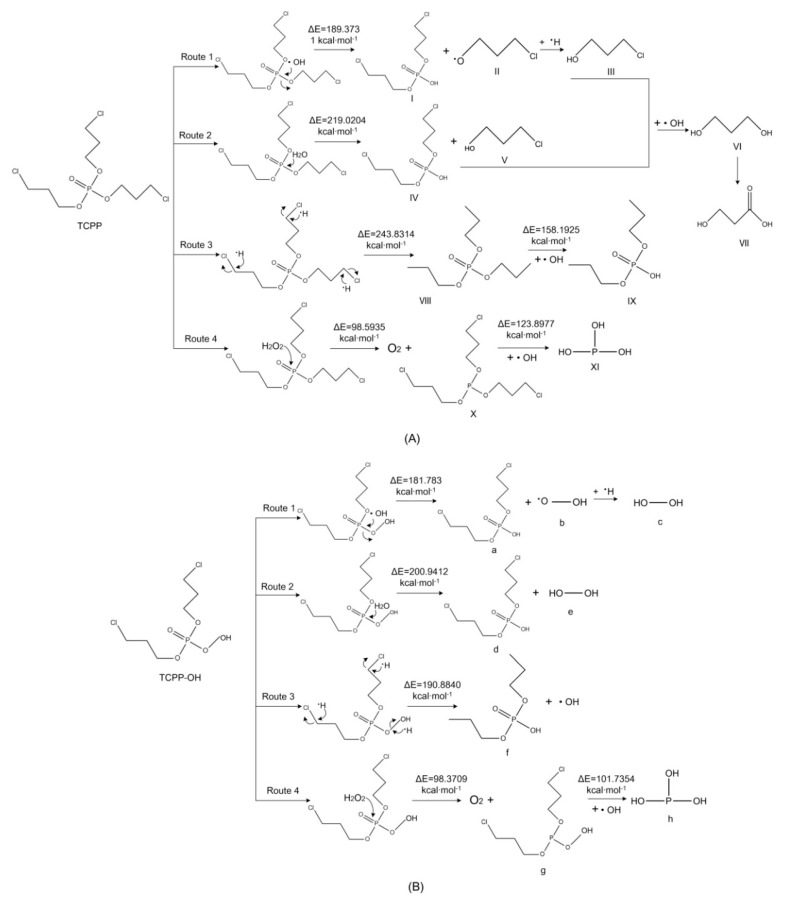
Proposed photodegradation pathways of TCPP before (**A**) and after modification (**B**).

**Figure 5 polymers-12-01672-f005:**
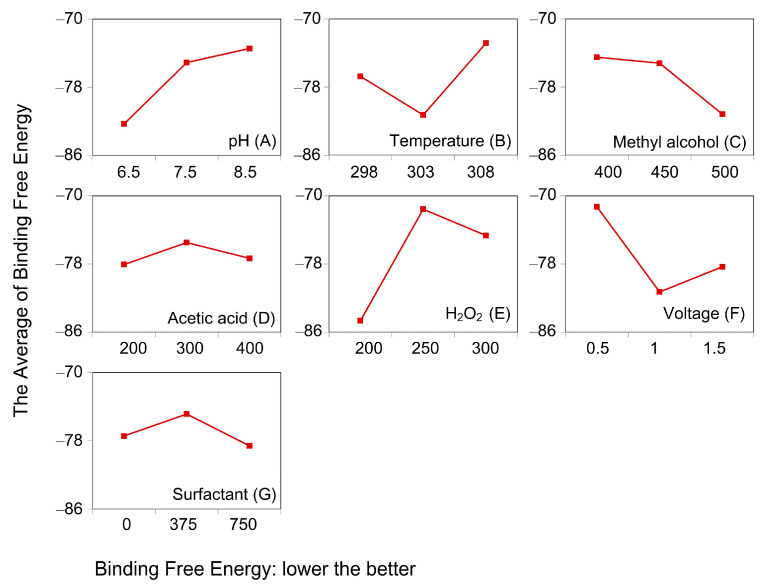
Main effect diagram for the average binding free energy value of degrading enzyme and TCPP–OH based on L_27_ (3^7^) Taguchi experimental design.

**Figure 6 polymers-12-01672-f006:**
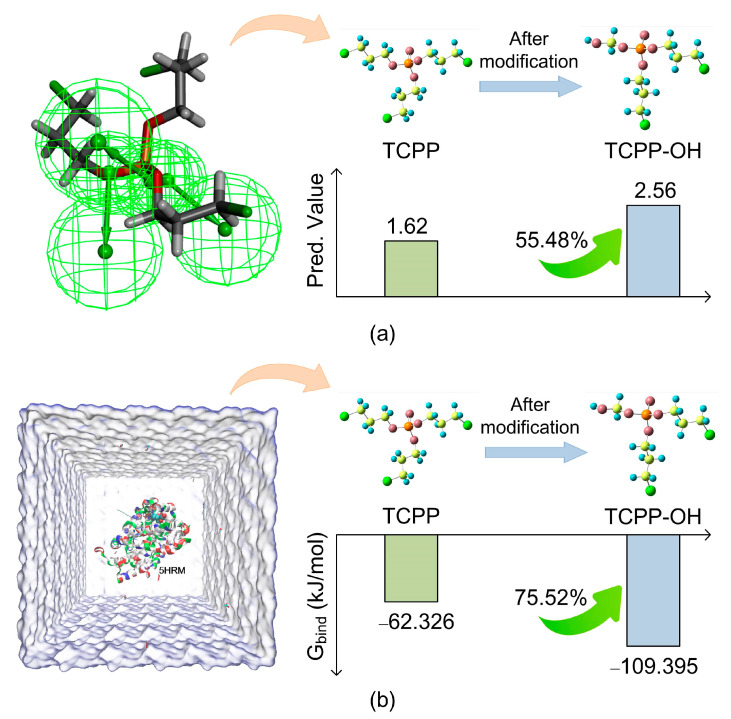
Diagram for the comparison of biodegradation efficiency of TCPP–OH using (**a**) biodegradation pharmacophore model prediction and (**b**) molecular dynamic simulation.

**Table 1 polymers-12-01672-t001:** Comprehensive effect evaluation values of biodegradation and photodegradation of organophosphorus flame retardants (OPFRs).

OPFRs	Biodegradation ΔG_bind_ (kJ/mol)	Efficacy Coefficient $R_{S1,i}$	Photodegradation Energy (eV)	Efficacy Coefficient $R_{S2,i}$	Comprehensive Evaluation Values C
TCEP ^a^	−123.731	1.168	7.2805	0.992	1.08
TCIPP ^a^	−144.537	1.000	7.2211	1.000	1.00
TCPP ^a^	−41.842	3.454	7.2874	0.991	2.22
TDCIPP	−144.5	1.000	7.1323	0.988	0.99
TDCPP ^a^	−203.57	1.408	7.1821	0.995	1.20
TPHP ^a^	−190.952	1.321	5.4695	0.757	1.04
EHDPP ^a^	−196.153	1.357	5.4015	0.748	1.05
TEP ^a^	−219.226	1.517	5.3858	0.746	1.13
TBOEP	−84.125	1.718	7.6896	0.939	1.33
TEHP ^a^	−268.708	1.859	8.2033	0.880	1.37
TPrP ^a^	−22.103	6.539	8.2739	0.873	3.71
BdPhP ^a^	−180.854	1.251	5.4003	0.748	1.00
TMP ^a^	−19.82	7.292	8.1823	0.883	4.09
TiBP ^a^	−148.43	1.027	8.1651	0.884	0.96
TPeP ^a^	−74.236	1.947	8.3083	0.869	1.41
TnPP ^a^	−9.191	15.726	8.1795	0.883	8.30
TmTP	−219.036	1.515	5.2731	0.730	1.12
TpTP	−259.422	1.795	5.2399	0.726	1.26
TBPP ^a^	−88.759	1.628	4.3292	0.600	1.11
TiPP	−112.943	1.280	8.1537	0.886	1.08
CDPP ^a^	−208.576	1.443	5.3779	0.745	1.09
IDPP ^a^	−255.722	1.769	5.3928	0.747	1.26

The full name of abbreviations of OPFRs are shown at Abbreviation Index; ^a^ represents the 17 molecules as training set to obtain the pharmacophore model for comprehensive biodegradation/photodegradation effects of OPFRs.

**Table 2 polymers-12-01672-t002:** Statistical parameters of the pharmacophore model for comprehensive biodegradation/photodegradation effects of OPFRs, constructed with Hypo-Gen.

Model	3D Space Relation of Hypo-Gen	Hypo No.	Total Cost	RMS	R^2^	Features
Pharmacophore model for comprehensive biodegradation/photodegradation effects	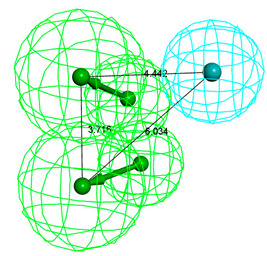	C1	70.55	0.24	0.90	HBA HBA H
		C2	75.06	0.00	0.00	HBA HBA H
		C3	75.65	0.83	0.88	HBA HBA H
		C4	75.06	0.00	0.00	HBA H
		C5	75.66	0.27	0.88	HBA HBA H
		C6	75.06	0.00	0.00	HBA HBA H
		C7	75.78	0.29	0.87	HBA HBA
		C8	75.06	0.00	0.00	HBA HBA
		C9	75.87	0.30	0.85	HBA HBA
	Configuration cost:	16.75	Fixed cost:	57.18	Null cost:	77.39

HBA—hydrogen bond acceptor; H—hydrophobic; HA—hydrophobic-aliphatic; RA—aromatic ring.

**Table 3 polymers-12-01672-t003:** Comprehensive evaluation value comparative results of HypoC1 and OPFR test sets.

	OPFRs	Fit Value	Estimate	Active	Error
Test set of comprehensive model HypoC1	TDCIPP	5.67	1.12	1.92	1.63
	TiPP	5.68	1.35	0.98	1.07
	TmTP	5.67	1.54	0.98	−1.28
	TpTP	5.64	1.65	1.34	−1.26
	TBOEP	5.65	1.45	1.22	1.13

**Table 4 polymers-12-01672-t004:** Optimal pharmacophore model parameters for biodegradation and photodegradation of OPFR molecules.

Model Name	Active	Training Set	Test Set	Configuration	R^2^	Total Cost	RMS
HypoB1	−log ΔG_bind_ (kJ/mol)	TPeP, TMP, TCPP, TnPP, TPrP, BdPhP, TDCPP, TPHP, TCIPP, TpTP	TBPP TiPP CDPP	13.24	0.77	48.17	0.03
HypoP1	Energy (eV)	TCEP, TCIPP, TCPP, TBOEP, TDCPP, TEHP, TPrP, TMP, TiBP, TPeP,	EHDPP TEP IDPP	14.29	0.82	50.23	0.20

**Table 5 polymers-12-01672-t005:** Predicted values and variation degree of comprehensive and single biodegradation/photodegradation effect models of OPFR derivatives.

	Compound	C_pred_	Fit Value	Reduction Intensity (%)	B_pred_ −log ΔG_bind_	Increased Intensity (%)	P_pred_ (eV)	Reduction Intensity (%)	Ratio
Before modification	TCPP	2.22	-	-	1.62	-	7.29	-	-
After modification	TCPP–OH	1.40	5.75	37.03%	2.52	55.48%	3.91	46.37%	1.20
	TCPP–CH_2_OH	1.24	5.82	44.06%	2.31	42.43%	3.43	52.98%	0.80
	TCPP–NH_2_	1.31	5.93	41.14%	2.57	58.76%	4.60	36.93%	1.59
	TCPP–COOH	1.49	5.84	32.94%	2.23	37.69%	4.35	40.38%	0.93
	TCPP–CONH_2_	1.38	5.81	37.90%	2.46	51.89%	4.49	38.37%	1.35
	TCPP–CHO	1.52	5.88	31.45%	2.17	34.12%	4.03	44.75%	0.76
	TCPP–PO_3_H_2_	1.39	5.85	37.38%	2.17	33.88%	4.02	44.88%	0.75
	TCPP–COOCH_3_	1.37	5.84	38.42%	2.58	58.82%	4.59	37.02%	1.59
	TCPP–SO_3_H	1.39	5.85	37.30%	2.17	33.77%	4.67	35.98%	0.94
Before modification	TCEP	1.08	-	-	2.09	-	7.28	-	-
After modification	TCEP–OH	1.01	5.84	6.36%	2.89	38.07%	4.50	38.25%	1.00
	TCEP–CH_2_OH	0.90	5.87	16.70%	3.06	46.38%	3.57	50.99%	0.91
	TCEP–NH_2_	0.71	5.89	26.55%	2.72	29.90%	3.51	51.76%	0.58
	TCEP–COOH	0.86	5.92	20.35%	2.97	42.15%	4.09	43.85%	0.96
	TCEP–CONH_2_	0.79	5.87	27.11%	3.00	43.60%	4.53	37.77%	1.15
	TCEP–CHO	0.61	5.93	26.40%	2.82	34.61%	3.46	52.41%	0.66
	TCEP–PO_3_H_2_	1.01	5.97	6.46%	3.01	43.98%	4.95	31.99%	1.37
	TCEP–COOCH_3_	0.79	5.79	26.87%	2.92	39.46%	5.05	30.58%	1.29
	TCEP–SO_3_H	0.85	5.91	21.08%	3.11	48.68%	3.78	48.03%	1.01

C_pred_, B_pred_ and P_pred_ represent the predicted values for comprehensive biodegradation/photodegradation effect, biodegradation effect and photodegradation effect pharmacophore models.

**Table 6 polymers-12-01672-t006:** Date statistics of flame retardancy, energy value, insulation and frequency of 14 OPFR molecules.

OPFR Derivatives		Flame Retardancy (kcal/mol)		Energy Gap (e.v.)		Energy (a.u.)		Freq
		Value	Enhanced Rate	Value	Change Rate (%)	Value	Change Rate (%)	Value
Before modification	TCPP	225.72	-	5.38	-	−1023.57	-	6.07
After modification	TCPP–OH	209.57	7.15%	5.15	−4.28%	−1536.08	−50.07%	2.34
	TCPP–CH_2_OH	256.70	−13.73%	5.56	3.35%	−1626.53	−58.91%	2.16
	TCPP–COOH	294.37	−30.42%	5.42	0.74%	−1695.34	−65.63%	2.44
	TCPP–CONH_2_	205.82	8.81%	5.31	−1.30%	−1563.15	−52.72%	2.70
	TCPP–CHO	215.94	4.33%	5.68	5.58%	−1532.77	−49.75%	2.51
	TCPP–PO_3_H_2_	201.83	10.58%	5.39	0.19%	−1506.84	−47.21%	3.08
	TCPP–SO_3_H	266.90	−18.24%	5.18	−3.72%	−1551.22	−51.55%	2.49
Before modification	TCEP	273.03	-	6.64	-	−1375.51	-	7.31
After modification	TCEP–OH	310.61	−13.76%	8.29	24.85%	−1475.55	−7.27%	5.89
	TCEP–CH_2_OH	322.95	−18.28%	8.9	34.04%	−1395.67	−1.47%	4.67
	TCEP–COOH	327.24	−19.85%	8.14	22.59%	−1402.43	−1.96%	5.14
	TCEP–CONH_2_	291.65	−6.82%	7.86	18.37%	−1433.19	−4.19%	5.28
	TCEP–PO_3_H_2_	238.74	12.56%	8.03	20.93%	−1452.28	−5.58%	4.72
	TCEP–COOCH_3_	255.25	6.51%	8.42	26.81%	−1466.84	−6.64%	5.49
	TCEP–SO_3_H	270.96	0.76%	8.34	25.60%	−1467.63	−6.70%	5.64

**Table 7 polymers-12-01672-t007:** Effect of the combination of external conditions on the biodegradation system of degrading enzyme and TCPP–OH.

Experiment No.	pH	Temperature (K)	Methyl Alcohol (mg/L)	Acetic Acid (mg/L)	H_2_O_2_ (mg/L)	Voltage (V/m)	Surfactant (mg/L)	ΔG_bind_ (kJ/mol)
1	6.5	298	400	200	300	0.5	0	−88.461
2	6.5	298	400	200	350	1	375	−77.231
3	6.5	298	400	200	400	1.5	750	−76.508
4	6.5	303	450	300	300	0.5	0	−75.584
5	6.5	303	450	300	350	1	375	−84.364
6	6.5	303	450	300	400	1.5	750	−90.252
7	6.5	308	500	400	300	0.5	0	−90.39
8	6.5	308	500	400	350	1	375	−75.769
9	6.5	308	500	400	400	1.5	750	−82.256
10	7.5	298	450	400	300	1	750	−80.197
11	7.5	298	450	400	350	1.5	0	−69.418
12	7.5	298	450	400	400	0.5	375	−70.905
13	7.5	303	500	200	300	1	750	−103.79
14	7.5	303	500	200	350	1.5	0	−84.974
15	7.5	303	500	200	400	0.5	375	−65.508
16	7.5	308	400	300	300	1	750	−82.256
17	7.5	308	400	300	350	1.5	0	−60.265
18	7.5	308	400	300	400	0.5	375	−58.644
19	8.5	298	500	300	300	1.5	375	−78.007
20	8.5	298	500	300	350	0.5	750	−62.709
21	8.5	298	500	300	400	1	0	−87.278
22	8.5	303	400	400	300	1.5	375	−74.697
23	8.5	303	400	400	350	0.5	750	−67.101
24	8.5	303	400	400	400	1	0	−85.287
25	8.5	308	450	200	300	1.5	375	−88.545
26	8.5	308	450	200	350	0.5	750	−62.199
27	8.5	308	450	200	400	1	0	−55.286

**Table 8 polymers-12-01672-t008:** Ranking result for the external conditions of biodegradation of degrading enzyme and TCPP–OH molecule based on L_27_ (3^7^) Taguchi experiment design.

Factors/Levels	A	B	C	D	E	F	G
1	−82.31	−76.75	−74.49	−78.06	−84.66	−71.28	−77.44
2	−75.11	−81.28	−75.19	−75.48	−71.56	−81.27	−74.85
3	−73.46	−72.85	−81.19	−77.34	−74.66	−78.32	−78.59
Ranking	3	4	5	7	1	2	6

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
