# Peer review of "Enhanced Biodegradation/Photodegradation of Organophosphorus Fire Retardant Using an Integrated Method of Modified Pharmacophore Model with Molecular Dynamics and Polarizable Continuum Model"

_polymers, 2020, doi:10.3390/polym12081672_

Round 1

Reviewer 1 Report

  1. Page 21 and Lane 217 there is mistake in the presenting data. Total cost data should be 70.55 and not 75.55. Please correct it.
  2. Authors have presented nice 3d space relation of Hypo Gen model. In this model author did not explain about distances between the 3d spears such 4.442, 3.715 and 5.034. Please explain these values and make it larger for visibility.
  3. It is also necessary to give some explanation for the validation and stability of this model.
  4. In Table 4. Line 244 and 245, Authors have mentioned that ‘minimum RMS error
    245 (0.03 and 2.20)’ but for the RMS errors should be below 2.00. Please explain this.
  5. In Figure 2, it is reasonable to mark all necessary degradation substitution positions instead of represented ones. (eg. In TCPP 4, 4’, and 4’’). If not please give reason behind this.

Author Response

We are very grateful for the reviewer’s comment. We carefully considered the reviewer's comments or recommendations and addressed them point-by-point. All the revised sections have been highlighted by using the "Track Changes" function in Microsoft Word in the revised manuscript and the detailed responses to the comments are listed in the response to reviewers.

Reviewer 2 Report

Authors provide a mechanicistic study on P-based flame retedardancy. This filed is dominated by only empirical approaches but this paper move a step forward proposing a solid and comprehensive study of flame retadancy using a molecular approach.

The main weak point of this paper is difficult of readers to orietatting himself though the plethora of acronyms used in the text.

Additionally:

lines 14-47: The abstract is far too long. Be more concise and reduce the length. Furthremore, do not use acronyms in the abstract.

line 76: if you use and acronym (TCEP, TCIPP, TDCPP) you must define it teh first time that you are using it.

Nonetheless, i warmly recommand teh pubblication of this paper in Polymers.

Author Response

(The authors gave the same response as above.)
